# *Salmonella* Typhimurium with Eight Tandem Copies of *bla*_NDM-1_ on a HI2 Plasmid

**DOI:** 10.3390/microorganisms12010020

**Published:** 2023-12-22

**Authors:** Huijuan Song, Siyu Zou, Yi Huang, Cui Jian, Weiyong Liu, Lei Tian, Lu Gong, Zhongju Chen, Ziyong Sun, Yue Wang

**Affiliations:** Department of Laboratory Medicine, Tongji Hospital, Tongji Medical College, Huazhong University of Science and Technology, Wuhan 430030, China; shj18134682591@163.com (H.S.); m202276182@hust.edu.cn (S.Z.); rookie7826@163.com (Y.H.); jiancui_tj@126.com (C.J.); wyliu@hust.edu.cn (W.L.); iso15189@126.com (L.T.); tjhgongl@163.com (L.G.); hailong1228@163.com (Z.C.); beiyue000@163.com (Y.W.)

**Keywords:** NDM-1, *Salmonella* Typhimurium, IncHI2, whole-genome sequencing, PacBio HiFi sequencing, tandem copies, carbapenem resistance

## Abstract

Carbapenem-resistant *Salmonella* has recently aroused increasing attention. In this study, a total of four sequence type 36 *Salmonella enterica* subsp. *enterica* serovar Typhimurium (*S.* Typhimurium) isolates were consecutively isolated from an 11-month-old female patient with a gastrointestinal infection, of which one was sensitive to carbapenems and three were resistant to carbapenems. Via antibiotic susceptibility testing, a carbapenemases screening test, plasmid conjugation experiments, Illumina short-reads, and PacBio HiFi sequencing, we found that all four *S.* Typhimurium isolates contained a *bla*_CTX-M-14_-positive IncI1 plasmid. One carbapenem-sensitive *S.* Typhimurium isolate then obtained an IncHI2 plasmid carrying *bla*_NDM-1_ and an IncP plasmid without any resistance genes during the disease progression. The *bla*_NDM-1_ gene was located on a new 30 kb multiple drug resistance region, which is flanked by IS*26* and TnAs2, respectively. In addition, the ST_F0903R isolate contained eight tandem copies of the IS*CR1* unit (IS*CR1*-*dsbD*-*trpF*-*ble*-*bla*_NDM-1_-IS*Aba125*Δ1), but an increase in MICs to carbapenems was not observed. Our work further provided evidence of the rapid spread and amplification of *bla*_NDM-1_ through plasmid. Prompting the recognition of carbapenem-resistant *Enterobacterales* and the initiation of appropriate infection control measures are essential to avoid the spread of these organisms.

## 1. Introduction

*Salmonella* is the leading cause of foodborne illness worldwide and is classified into various serovars by the White–Kauffmann–le Minor scheme based on its surface antigenic composition [1]. There are a variety of clinical manifestations associated with non-typhoidal serovars of *Salmonella enterica* (NTS), but the most frequently observed symptom is self-limiting gastroenteritis. In general, empiric antimicrobial therapy is not recommended unless a disseminated infection occurs or infection occurs in immunocompromised patients. The empiric antimicrobial therapy includes a third-generation cephalosporin or azithromycin for children [2]. However, with the abuse of antibiotics, the drug resistance of *Salmonella* has become increasingly serious and poses a serious threat to public health [3,4]. Recently, the emergence of carbapenem-resistant *Salmonella* has raised international attention, although it has rarely been reported [5]. While addressing the therapeutic challenges posed by carbapenem-resistant *Salmonella*, it is crucial to understand its resistance mechanisms and take appropriate measures to control the spread of resistance.

NTS has been reported to have acquired carbapenem resistance through the acquisition of carbapenemase genes which produce carbapenemases such as KPC [6], NDM [7], IMP [8], VIM [9], and OXA-48 [10]. Since it was first reported in 2009 [11], the *bla*_NDM-1_ gene-positive *Enterobacterales* has spread rapidly around the world [12]. The *bla*_NDM_ genes were usually reported to be located on the IncX3, IncC, IncL, IncM, and IncN plasmids [12]. The IncHI2 plasmid-carrying *bla*_NDM_ gene is rarely reported [12,13,14]. Recent research has shown that mobile genetic elements (MGEs) play a crucial role in facilitating the rapid transmission of *bla*_NDM_ genes [15,16]. Such MGEs, like IS*26* and IS*CR1*, which are frequently found in the vicinity of *bla*_NDM_ genes among diverse strains, most likely contribute to the dissemination of *bla*_NDM_ genes [17,18]. Currently, there have been two reported cases of *Salmonella enterica* subsp*. enterica* serovar Typhimurium (*S.* Typhimurium) carrying the *bla*_NDM-1_ gene. Banerjee et al. reported the first case of *S.* Typhimurium harboring *bla*_NDM-1_ isolated from burn wounds [19]. In 2019, an *S.* Typhimurium isolate carrying *bla*_NDM-1_ was found in 241 children infected with gastroenteritis due to NTS from three public hospitals in Hong Kong [20]. The location and genetic environment of *bla*_NDM-1_ were not further studied in the above *S.* Typhimurium isolates.

Herein, our investigation focused on the transformation of carbapenem resistance in four continuously isolated *S.* Typhimurium isolates from a patient who developed diarrhea. Using phenotypic experiments, whole genome sequencing, and in-depth genomic analysis, we have clarified the antibiotic resistance characteristics and the mechanism of resistance to carbapenems within *S.* Typhimurium. Moreover, we described, in detail, the genetic environment and transmission mechanism of the *bla*_NDM-1_ gene, which provided a theoretical basis for clinical prevention, as well as research data for the epidemiological investigation of carbapenem-resistant *Enterobacterales* worldwide.

## 2. Materials and Methods

### 2.1. Bacterial Isolates and Case Information

A total of four isolates were used in this study and were derived from fecal specimens from different dates obtained from an 11-month-old patient who was admitted to Tongji Hospital: 29 August (ST_F0829S), 3 September (ST_F0903R), 7 September (ST_F0907R), and 13 September 2020 (ST_F0913R). The patient presented with symptoms of diarrhea (4–5 times/day) and fever upon admission. Cefoperazone-tazobactam was administered on 2 September, but the patient developed a wind-like rash on the face after treatment. After oral administration of loratadine on 3 September, the patient’s rash subsided, indicating a possible allergic reaction to Cefoperazone-tazobactam. Consequently, the medication was switched to meropenem on 4 September. However, on 10 September, the patient’s diarrhea symptoms did not significantly improve. Therefore, meropenem was discontinued, and azithromycin was administered instead. After the azithromycin treatment, the patient’s condition improved.

### 2.2. Strain Identification and Sequence Typing

The strain was identified by matrix-assisted laser desorption/ionization time-of-flight mass spectrometry (MALDI-TOF MS) (Autof ms1000, Zhengzhou, China), and then the serotype was identified using the local Centers for Disease Control and Prevention (CDC, Atlanta, GA, USA) using Statens Serum Institut (SSI, Copenhagen, Denmark) serotyping reagent. Multilocus sequence typing (MLST) was further performed by polymerase chain reaction (PCR) with primers designed according to the EnteroBase database. Sequence types (STs) were identified by aligning Sanger sequencing results of amplicons with the EnteroBase database (https://enterobase.warwick.ac.uk/species/index/senterica accessed on 26 March 2022).

### 2.3. Antibiotic Susceptibility Testing

We performed antibiotic susceptibility testing using broth microdilution method. For imipenem–relebactam, aztreonam–avibactam, colistin, eravacycline, and tigecycline, the breakpoints specified by the European Committee on Antimicrobial Susceptibility Testing (EUCAST, Växjö, Sweden) were used [21]. Imipenem, meropenem, meropenem-vaborbactam, ceftazidime, ceftazidime–avibactam, polymyxin, amikacin, cefepime, aztreonam, ciprofloxacin, and trimethoprim–sulfamethoxazole were interpreted according to the standards of the Clinical and Laboratory Standards Institute (CLSI) M100 ED33 guideline [22]. *Escherichia coli* (*E. coli*) ATCC 25922 and *Pseudomonas aeruginosa* ATCC 27853 were used as quality control standards [21,22].

### 2.4. Screening for Carbapenemases

The modified carbapenem inactivation method (mCIM) and EDTA-carbapenem inactivation method (eCIM) were used to detect carbapenemases [23]. Carbapenemase types (IMP, KPC, VIM, OXA-48 family, and NDM) were further investigated using an NG–Test Carba 5 assay (NG Biotech, Guipry, France). Additionally, PCR was performed to detect five common carbapenemase genes (*bla*_KPC_, *bla*_IMP_, *bla*_VIM_, *bla*_NDM_, and *bla*_OXA-48_) with primers as described previously [24]. We aligned the sequencing results of the amplicons using BLAST (http://www.ncbi.nlm.nih.gov/BLAST accessed on 26 March 2022).

### 2.5. Conjugation Assay and PCR-Based Replicon Typing (PBRT)

The transferability of carbapenem resistance was further investigated with plasmid conjugation experiments. The rifampin-resistant *E. coli* isolate C600 was used as the recipient and 2 mg/L meropenem plus 600 mg/L rifampin was used for selecting transconjugants on Luria–Bertani (LB) agar plates. Plasmid replicon types were identified by PBRT with primers and procedures as described previously [25].

### 2.6. Pulsed-Field Gel Electrophoresis (PFGE)

The genomic DNA from isolates prepared in agarose blocks was digested with restriction enzymes XbaI (Takara Bio, Otsu, Japan) at 37 °C for 4 h or S1 nuclease (Takara Bio, Otsu, Japan) at 23 °C for 40 min. PFGE was performed using the CHEF-Mapper XA System (Bio-Rad, Hercules, CA, USA) at 6.0 V/cm and 14 °C (switch time: 2.16–63.80 s) for 18.5 h.

### 2.7. Illumina Short Reads Sequencing and Assembly

Bacterial genomic DNA was extracted from overnight cultures with LB broth by the SDS method [26]. The purity and integrity of the extracted DNA were assessed with 1% agarose gel electrophoresis, and the concentration was quantified using a Qubit^®^ 3.0 fluorometer (Thermo Scientific, Waltham, MA, USA). The 350 bp sequencing libraries were generated using NEBNext^®^ Ultra™ DNA Library Prep Kit for Illumina (NEB, Ipswich, MA, USA) following the manufacturer’s recommendations. Libraries were analyzed for size distribution by Agilent2100 Bioanalyzer and quantified using real-time PCR. The whole genome was sequenced using Illumina NovaSeq 6000 (PE150). The reads containing adapter, host contamination, and too many low-quality or N bases, were removed from Illumina sequencing raw data to obtain clean data. The assembly was performed by SOAPdenovo (https://sourceforge.net/projects/soapdenovo2/ accessed on 20 June 2022, v2.04), SPAdes (https://cab.spbu.ru/software/spades/ accessed on 20 June 2022, v3.14.0), and Abyss (http://www.bcgsc.ca/platform/bioinfo/software/abyss accessed on 20 June 2022, v2.0.2) software using the optimal k-mer value. The assembly results from the three software were integrated and optimized using CISA (http://sb.nhri.org.tw/CISA/en/CISA accessed on 20 June 2022) software. The gaps were filled using the GapCloser (https://sourceforge.net/projects/soapdenovo2/files/GapCloser/ accessed on 20 June 2022) software to obtain the final draft genome.

### 2.8. Pacific Biosciences (PacBio) High-Accuracy Long-Read (HiFi) Sequencing and Assembly

The PacBio HiFi sequencing was performed as previously described [27,28]. Bacterial genomic DNA was extracted using the SDS method, as previously described [26]. The extracted DNA was detected by 1% agarose gel electrophoresis (180 V, 20 min) and 0.8% PFGE (5–80 kb, 17 h) and quantified by Qubit. The 10–20 kb libraries were prepared using the SMRTbell™ Express Template Prep Kit 2.0 following the manufacturer’s recommendations and quantified by Qubit. The size of the inserted fragment was detected with an Agilent2100 Bioanalyzer. The constructed library complexes were then sequenced on the PacBio Sequel II Sequencing platform using Circular Consensus Sequencing (CCS) mode to generate HiFi reads by the SMRT Link (https://www.pacb.com/support/software-downloads/ accessed on 10 June 2023, v8.0). HiFi reads were assembled using the Hifiasm-0.19.5-r578 (https://github.com/chhylp123/hifiasm accessed on 10 June 2023) to produce the complete genome.

### 2.9. Genome Annotation and Comparative Genomics Analysis

BLAST (https://blast.ncbi.nlm.nih.gov/Blast.cgi accessed on 10 July 2023) was used to identify sequences with the highest similarity to the chromosome and plasmid. We identified plasmid incompatibility group (Inc.) types using PlasmidFinder (https://cge.food.dtu.dk/services/PlasmidFinder/ accessed on 10 July 2023). The genome was annotated using Prokka (https://github.com/tseemann/prokka accessed on 10 July 2023, v1.14.6) and RAST (https://rast.nmpdr.org/ accessed on 10 July 2023). Antimicrobial resistance genes (ARGs) in the genome were identified using the ABRicate program (https://github.com/tseemann/abricate accessed on 10 July 2023, v1.0.1) based on the Comprehensive Antibiotic Resistance Database (CARD) (https://card.mcmaster.ca/analyze accessed on 10 July 2023) and ResFinder Database (https://cge.cbs.dtu.dk/services/ResFinder/ accessed on 10 July 2023). ISfinder (https://www-is.biotoul.fr/blast.php accessed on 10 July 2023) was used to insert sequence (IS) annotations. Easyfig (http://mjsull.github.io/Easyfig/ accessed on 10 July 2023, v2.2.5) and BRIG (https://sourceforge.net/projects/brig/ accessed on 15 July 2023, v0.95) was used for comparative genomic analysis and generating maps. BWA (https://bio-bwa.sourceforge.net/ accessed on 15 July 2023, v0.7.12) and samtools (https://samtools.sourceforge.net/ accessed on 15 July 2023, v1.7) were employed to calculate Illumina sequencing reads mapping depths and then plotted out in R (https://www.r-project.org/ accessed on 15 July 2023, v4.3.0).

### 2.10. Real-Time Quantitative PCR (qRT-PCR)

Copy numbers and expression levels of the *bla*_NDM-1_ and the IncHI-type plasmid replication initiator gene *repHI2* were determined using qRT-PCR. Total RNA was extracted using an RNA Extraction Kit (Promega, Milan, Italy). Reverse transcription was performed using the PrimeScript^TM^ RT Reagent Kit (Takara Bio, Beijing, China). Finally, the quantitative PCR was performed using SYBR qPCR Master Mix (Vazyme, Nanjing, China) and LightCycler 480 (Roche, Basel, Switzerland). The *16Sr RNA* was used as the reference gene [29]. The qRT-PCR analysis was performed using three biological replicates and three technical replicates. The copy numbers and expression levels were calculated using the 2^−ΔΔCt^ method. The independent sample *t*-test was used for statistical analysis, and a *p* value < 0.05 was regarded as statistically significant. The *repHI2*, *16S rRNA* and *bla*_NDM-1_ primers used in this study were: q-HI2-F (GGATGGGCATCATTCGAACC), q-HI2-R (GGTGAACGACAAGGTAACGG), q-16S-F (CATCATGGCCCTTACGACCAG), q-16S-R (ACGATTACTAGCGATTCCGACT), q-NDM-F (TTTGGCGATCTGGTTTTCCG), and q-NDM-R (ATCAAACCGTTGGAAGCGAC), respectively.

## 3. Results

### 3.1. Strain Identification

Four isolates were identified as ST36 *S.* Typhimurium. The PFGE results for the four isolates of *S.* Typhimurium are shown in Appendix A. The XbaI-PFGE band profiles of these isolates exhibited a high degree of similarity, and therefore represent the same strain.

### 3.2. Phenotypic and Genotypic Characteristics of the Strain

The minimum inhibitory concentrations (MICs) of *S.* Typhimurium isolates to antibiotics are shown in Table 1. The results showed that the first isolate (ST_F0829S) of *S*. Typhimurium from the patient was sensitive to carbapenems (including imipenem, imipenem-relebactam, meropenem, and meropenem-vaborbactam), while ST_F0903R, ST_F0907R, and ST_F0913R were all resistant to the above carbapenems. The ST_F0903R, ST_F0907R, and ST_F0913R exhibited resistance to third-generation cephalosporins but were sensitive to quinolones. The MICs of ST_F0903R, ST_F0907R, and ST_F0913R isolates to imipenem and imipenem–relebactam have no differences, both of which are 16 mg/L. The MICs of the ST_F0907R isolate to meropenem and meropenem-vaborbactam were slightly higher than that of the ST_F0903R and ST_F0913R isolates. The ST_F0903R, ST_F0907R, and ST_F0913R isolates were confirmed to be positive in both the mCIM and eCIM tests. Additionally, these isolates were NDM-positive in the NG-Test Carba 5 assay. The PCR and sequencing results of carbapenemase genes were consistent with the *bla*_NDM-1_ gene.

### 3.3. Identification of Conjugative Resistance Plasmid

The S1-PFGE band profiles showed that both the ST_F0907R and ST_F0913R isolates exhibited the presence of approximately 90 kb, 300 kb, and 50 kb circular plasmids. In contrast, the ST_F0903R isolate lacked the ~50 kb circular plasmid, while the ST_F0829S isolate only possessed a ~90 kb circular plasmid. Further, the PBRT results revealed that ST_F0829S carried only one kind of plasmid replicon IncI1, ST_F0903R harbored two kinds of plasmid replicons, IncI1 and IncHI2, while ST_F0907R and ST_F0913R possessed three kinds, IncI1, IncHI2, and IncP. Carbapenem resistance could be transferred from ST_F0903R, ST_F0907R, and ST_F0913R isolates to recipient *E. coli* C600 by conjugation. PCR amplification and Sanger sequencing results confirmed that the transconjugants (ST_F0903Rtrans, ST_F0907Rtrans, and ST_F0913Rtrans) obtained both the *bla*_NDM-1_ gene and IncHI2 plasmid replicon. The MICs of three transconjugants were shown in Table 1. These results indicated that the *bla*_NDM-1_ gene in ST_F0903R, ST_F0907R, and ST_F0913R isolates was located on about 300 kb transferable IncHI2 plasmid.

### 3.4. The ARGs and Plasmid Replicons of the Four S. Typhimurium Isolates

The distribution of ARGs and plasmid replicons in draft genomes of the four *S*. Typhimurium isolates (ST_F0829S, NZ_JAVSJC000000000.1; ST_F0903R, NZ_JAVSJD000000000.1; ST_F0907R, NZ_JAVSJE000000000.1; ST_F0913R, NZ_JAVSJF000000000.1) are shown in Figure 1. The genome annotation results showed that the plasmid replicon types in the four isolates were consistent with the findings obtained from the PBRT experiments. By aligning the sequences containing the IncP1 replicon in contigs of ST_0907R, and ST_0913R isolates, we obtained the complete sequence of the circular IncP1 plasmid named pST_P1 (Appendix A). This plasmid had a size of 55,072 bp and an average GC content of 46%, which did not encode any ARGs. The ST_F0829S isolate carried only the *aac*(*6′*)*-Iaa* and *bla*_CTX-M-14_ genes. Among the four isolates of *S*. Typhimurium, there were no mutations in the *aac*(*6′*)*-Iaa* gene, which was consistent with the sensitivity of amikacin. The ARGs in the three carbapenem-resistant *S*. Typhimurium isolates (ST_F0903R, ST_F0907R, and ST_F0913R) were found to be completely identical, including *bla*_CTX-M-14_, *bla*_NDM-1_, *ble*, *sul1*, *aadA5*, *dfrA17*, *msr*(*E*), *mph*(*E*), and *tet*(*B*), providing resistance to cephalosporins, carbapenems, bleomycin, sulfonamides, aminoglycosides, trimethoprim, macrolides, and tetracycline, respectively (Figure 1).

### 3.5. The Features of the Chromosome and Plasmids of the ST_F0903R Isolate

The initially isolated carbapenem-resistant *S*. Typhimurium (ST_F0903R) was further subjected to PacBio HiFi sequencing to obtain the complete genome sequences of this isolate. Through the assembly of whole-genome sequencing data, we characterized the circular chromosome and two circular plasmids of the ST_F0903R isolate. The ST_F0903R chromosome (CP129630) carried the *aac*(*6′*)*-Iaa* gene, with a size of 4770057 bp and an average of 52.13% GC content (Appendix A). We further mapped the contigs of the four *S*. Typhimurium isolates to the chromosome and plasmids of the ST_F0903R isolate. The mapping results demonstrated that the chromosome sequences of the four isolates were almost identical, suggesting that they may have originated from a single parental clone (Appendix A).

In the ST_F0903R isolate, one of the plasmids we identified was a 98748 bp IncI1 plasmid (pST_I1_CTX-M-14, CP129632) with a GC content of 49.95%, which only carried the *bla*_CTX-M-14_ gene (Appendix A). From the contigs mapping, pST_I1_CTX-M-14 was also present in the ST_0907R, ST_0913R, and ST_F0829S isolates (Appendix A). The results from BLAST alignment indicated that pST_I1_CTX-M-14 was quite common and almost identical to *E. coli* isolate 105CF plasmid p105CF (GenBank accession no. MK764025.1) isolated from Japanese beef cattle in 2016 (Appendix A).

The other plasmid identified was a 321025 bp plasmid with an average G + C content of 47% named pST_HI2_NDM-1 (CP129631), which carried both the IncHI2 and IncHI2A replicons and encoded a wide variety of ARGs (*bla*_NDM-1_, *ble*, *sul1*, *aadA5*, *dfrA17*, *msr*(*E*), *mph*(*E*), and *tet*(*B*)) (Figure 2). From the contigs mapping, pST_HI2_NDM-1 was also found to be present in ST_0907R and ST_0913R, but not ST_F0829S (Figure 2). In summary, initially, the carbapenem-sensitive *S.* Typhimurium carried a *bla*_CTX-M-14_-positive IncI1 plasmid. Over time, this isolate acquired an IncHI2 plasmid carrying the *bla*_NDM-1_ gene, as well as an IncP plasmid that did not possess any ARGs. The medical history of the patient and the plasmid transfer events in the *S.* Typhimurium isolate are shown in Figure 3. By BLAST aligning, we found that the backbone of pST_HI2_NDM-1 was closely related to previously identified IncHI2 plasmids such as p7926H (MZ750395.1), p49589CZ_VIM (CP085773.1), pSL131_IncHI2 (MH105051.1), p7994H (MZ855470.1), and pMY460-rmtE (LC511997.1), but was not completely identical due to differences in mobility elements, the surrounding resistance genes, and hypothetical proteins (Figure 2). One of them, p49589CZ_VIM, had the highest level of similarity in the plasmid backbone region, carrying the IncHI2/2A replicons but lacking the *bla*_NDM-1_ gene. It was derived from *Enterobacter hormaechei* and isolated from a patient’s decubitus swab from Prague, the Czech Republic, in 2019 [30]. Among the aforementioned plasmids, only p7926H carried the *bla*_NDM-1_ gene, derived from *Enterobacter hormaechei* subsp. *Steigerwaltii*, which was isolated from Warsaw, Poland, in 2017 [31]. However, the genetic context of the *bla*_NDM-1_ gene of this plasmid differed from that of pST_HI2_NDM-1.

### 3.6. Genetic Context of the bla_NDM-1_ Gene

The alignment results for the *bla*_NDM-1_ gene in the draft genomes showed that the genetic context of the *bla*_NDM-1_ gene carried by three carbapenem-resistant *S*. Typhimurium isolates (ST_F0903R, ST_F0907R, and ST_F0913R) was completely consistent (Figure 4A). In pST_HI2_NDM-1, the *bla*_NDM-1_ gene was located in a single multidrug resistance (MDR) region of approximately 30 kb, bracketed by downstream IS*26* and upstream Tn*As2,* and the entire region was named Tn*As2*-MDR-IS*26* (Figure 4B). In addition to the *bla*_NDM-1_ gene, several genes (*merT-merP-merC-merA-merD-merE*) related to mercury resistance and other ARGs (*dfrA17*, *aadA5*, *sul1*, *ble*, *msr(E)*, and *mph(E)*) were also identified in this region (Figure 4B). The *bla*_NDM-1_ gene was located downstream of a class 1 integron harboring a cassette array (*dfrA17*-*aadA5*) and embedded the structure (*dsbD*-*trpF*-*ble*-*bla*_NDM-1_-IS*Aba125*Δ1) between two IS*CR1* elements (Figure 4B). This structure, along with a single IS*CR1* element on one side, formed the 5654 bp IS*CR1* unit (Figure 4B). Searching in the NCBI Nucleotide collection (nt) database, it was observed that the sequence of the Tn*As2*-MDR-IS*26* region in pST_HI2_NDM-1 exhibited significant similarity to pNDM-MCR10 (CP135262.1) from *Enterobacter asburiae*, an unnamed plasmid (CP085197.1) from *Klebsiella quasipneumoniae* strain NDM-101, pKP-14-6-NDM-1 (MN175387) from *Klebsiella pneumoniae* strain, and p13ZX36-200 (MN101853.1) from *E. coli*. The main variations between the *bla*_NDM-1_-carrying regions in these plasmids were in the downstream sequence of *intI1*, the cassette array of the class 1 integron, and the ISs upstream of the *bla*_NDM-1_ gene (Figure 4B). Among the above plasmids, pNDM-MCR10 had the highest sequence similarity with the Tn*As2*-MDR-IS*26* region (Figure 4B). This plasmid was isolated in China in 2022 and contained IncFIB/FII replicons along with *bla*_NDM-1_ and *mcr-10* genes. The cassette array of the class 1 integron in this plasmid consisted of *ANT(2”)-Ia* and *aadA2*, with IS*1R* located upstream of *bla*_NDM-1_ instead of IS*Aba125* (Figure 4B). Apart from the *bla*_NDM-1_-harboring region described above, the remaining sequence of pNDM-MCR10 was different from pST_HI2_NDM-1.

### 3.7. Amplification of ISCR1 Unit Carrying bla_NDM-1_

Notably, the IS*CR1* unit carrying the *bla*_NDM-1_ gene was embedded in the pST_HI2_NDM-1 with eight tandem copies (Figure 2). Given that complete genome sequences of the ST_F0907R, and ST_F0913R isolates were not available, we mapped short Illumina sequencing reads to the Tn*As2*-MDR-IS*26* region harboring the *bla*_NDM-1_ gene to assess the copy number of the *bla*_NDM-1_ gene in ST_F903R, ST_F0907R, and ST_F0913R isolates. In the ST_F903R isolate, compared with other regions, the IS*CR1* unit containing the *bla*_NDM-1_ gene had significantly higher coverage of reads relative to the IncHI-type plasmid replication initiator gene *repHI2* and chromosome housekeeping gene *purE*, while in the ST_F0907R and ST_F0913R isolates, all genes in the TnAs2-MDR-IS*26* region had similar read depth ratios (Appendix A). This further suggested that there were tandem copies of the *bla*_NDM-1_ gene in the ST_F903R isolate, while in ST_F0907R and ST_F0913R, there was a single copy of the *bla*_NDM-1_ gene. The qRT-PCR results revealed that, when using the *16S rRNA* gene as a reference, there was no significant difference in the relative copy number of *bla*_NDM-1_ among the three isolates (Figure 5A). However, ST_F0907R exhibited a significantly higher relative expression level of *bla*_NDM-1_ compared to the other two isolates (Figure 5B). Moreover, the relative copy number and expression level of *repHI2* in ST_F0903R were significantly lower (Figure 5A,B).

## 4. Discussion

*Salmonella* infection is a significant global health concern, affecting approximately 150 million individuals worldwide and resulting in 60,000 deaths annually, according to the CDC. The issue of antimicrobial resistance in *Salmonella* has become increasingly concerning in recent years, as it exhibits varying degrees of resistance to multiple antibiotics, including fluoroquinolones and third-generation cephalosporins [3,4]. Carbapenems are considered the last resort for combating multidrug-resistant bacteria. However, the emergence and increase in carbapenem-resistant *Salmonella* make the management of *Salmonella* infections even more challenging. Therefore, it is crucial that we understand the mechanisms underlying carbapenem resistance in *Salmonella* to develop effective strategies for preventing and controlling the transmission of resistance.

Compared to the majority of studies that focus solely on single carbapenem-resistant isolates, our research involved the continuous isolation of both carbapenem-sensitive and carbapenem-resistant *S*. Typhimurium isolates from fecal samples from a single patient. Through comprehensive genomic studies, we were able to gain a more in-depth understanding of the transfer process of carbapenemase genes and plasmids in *Salmonella*. By employing state-of-the-art PacBio HiFi sequencing methods, we revealed the evolution process of drug resistance in *Salmonella* due to the acquisition of the HI2 plasmid carrying the *bla*_NDM-1_ gene, along with eight tandem copies of IS*CR1* unit (IS*CR1*-*dsbD*-*trpF*-*ble*-*bla*_NDM-1_-IS*Aba125*Δ1-*sul1*Δ1) on the HI2 plasmid. This sequencing method can provide more accurate and complete genome assembly results, which is of great significance for the study of complex genomes [32]. In China, *S.* Typhimurium is one of the main serotypes of NTS that cause human infection, often causing diarrhea in patients [33]. The *S.* Typhimurium isolates causing diarrhea in this study belonged to ST36, whereas the epidemic clone in NTS in China was ST19 and ST34 [20,34].

Our study revealed that the *bla*_NDM-1_ gene in pST_HI2_NDM-1 exhibited a distinct genetic context compared to the currently known isolates. Furthermore, the backbone of pST_HI2_NDM-1 showed a high level of similarity to, albeit not complete identity with, the known IncHI2 plasmids. These findings suggested that pST_HI2_NDM-1 may be an entirely new plasmid, and may have originated from multiple horizontal gene transfer events. Additionally, previous reports indicated that IncHI2 plasmids also carry other resistance genes, including *bla*_NDM-5_ [35], *bla*_VIM-1_ [36], and *bla*_MCR-1_ [37]. In this study, we did not find any carbapenemase genes other than *bla*_NDM-1_ on the plasmid. These findings highlighted the significant role played by IncHI2 plasmids in the dissemination of bacterial resistance. Furthermore, given the transferability of the IncHI2 plasmids, high vigilance should be exercised against such plasmids, which promote the widespread spread of drug resistance, and effective prevention and control measures are urgently needed.

Our study also identified the existence of tandem copies of the *bla*_NDM-1_ gene in the pST_HI2_NDM-1. To our knowledge, the structure of the IS*CR1* unit (IS*CR1*-*dsbD*-*trpF*-*ble*-*bla*_NDM-1_-IS*Aba125*Δ1) with the tandem copies in this study is different from that previously reported [38,39]. Two tandem copies of the IS*CR1* unit (*sul1*-*arr*-*3*-*cat*-*bla*_NDM-1_-*ble*-IS*CR1*) were reported in the *E. coli* Y5 chromosome (CP013483) in 2016 [18]. Although the reported genetic context of *bla*_NDM-1_ multiple copies is diverse, IS*CR1* is often found in the vicinity of the *bla*_NDM-1_ gene and is considered to be involved in the formation of *bla*_NDM-1_ tandem copies under the stress of carbapenems, which has not been verified by experiments [16]. Previous studies have established a model for the IS*CR1*-mediated amplification of the *qnrB2* gene via sequence alignment analysis and the rolling-circle transposition characteristics of IS*CR1* [40]. Based on this, we hypothesized that IS*CR1* formed a circular intermediate via an oriIS-mediated sequence (*dsbD-trpF-ble-bla*_NDM-1_-IS*Aba125*Δ*1*) that is directly inserted into a single copy of IS*CR1* at the end of the 3′-conserved segment (3′-CS), thereby mediating the amplification of the *bla*_NDM-1_ gene.

Previous studies have mainly reported the presence of multiple copies of the *bla*_NDM_ gene and their tandem copy structure. It is not common to isolate strains carrying single-copy and multiple-copy of the *bla*_NDM_ gene from the same patient. Therefore, there is limited research on the correlation between *bla*_NDM_ gene amplification and drug resistance level. At present, most studies have focused on the increased copy numbers in the *bla*_KPC_ gene, which can enhance drug resistance to ceftazidime-avibactam and meropenem-vaborbactam [41,42]. Additionally, previous studies have demonstrated that isolates carrying multiple copies of *bla*_VIM-1_ exhibit a onefold increase in MIC for carbapenems compared to isolates with a single copy [43]. However, the plasmid structures carrying *bla*_VIM-1_ and the genetic environments of *bla*_VIM-1_ are completely different between the two isolates [43]. Therefore, the increase in MIC values for carbapenems cannot be solely attributed to the tandem copy of the *bla*_VIM-1_ gene. In 2021, Simner et al. reported a case of resistance to cefiderocol in the presence of increased copy numbers and expression levels of the *bla*_NDM-5_ gene [44]. Due to the limitations of the concentration gradient of drugs, it was not feasible to compare the differences in MIC values of other cephalosporins or carbapenems [44]. Notably, our research findings indicated that the only difference in the IncHI2 plasmid among the three carbapenem-resistant *Salmonella* isolates was the copy number of the IS*CR1* unit carrying *bla*_NDM-1_. Despite the presence of multiple copies of *bla*_NDM-1_ on the IncHI2 plasmid in ST_F0903R, the copy number of this plasmid was lower compared to the other isolates. As a result, this did not lead to an increase in the expression level of *bla*_NDM-1_, thus explaining the similar levels of resistance observed in the three isolates.

In this study, we identified the presence of the *bla*_NDM-1_ gene and tandem copies of *bla*_NDM-1_ in the IncHI2 plasmid of *S*. Typhimurium isolates, highlighting the horizontal transmission mode of *bla*_NDM-1_ and the mechanism of IS*CR1*-mediated amplification within *S*. Typhimurium. However, there are certain limitations to this study. Firstly, the amplification of *bla*_NDM-1_ mediated by IS*CR1* has not been verified experimentally, and the specific mechanism has not been clarified. Secondly, further research is needed to investigate the relationship between the tandem copy of the *bla*_NDM-1_ gene and its expression level.

## 5. Conclusions

In conclusion, we elucidated the transfer process for plasmid and the *bla*_NDM-1_ gene in *S*. Typhimurium and the contribution of IS*CR1* during the amplification of the *bla*_NDM-1_ gene. Our study underscored the importance of promptly adjusting antibiotics for infection to control infections before bacterial resistance evolves. Given the potential for food-borne spread of carbapenem-resistant *Salmonella*, surveillance of these isolates should be strengthened.

## Figures and Tables

**Figure 1 microorganisms-12-00020-f001:**
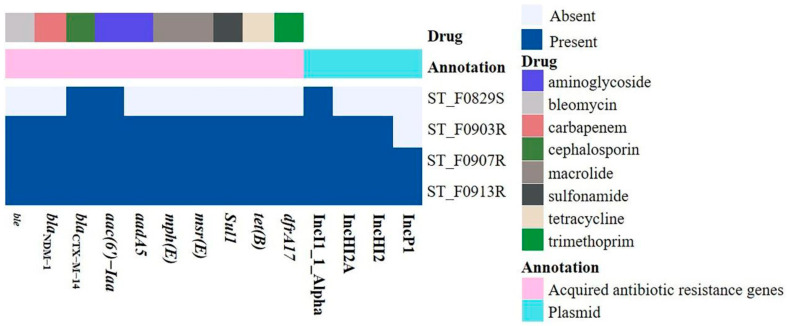
The distribution of antibiotic resistance genes and plasmid replicons in draft genomes of the four *Salmonella* Typhimurium isolates.

**Figure 2 microorganisms-12-00020-f002:**
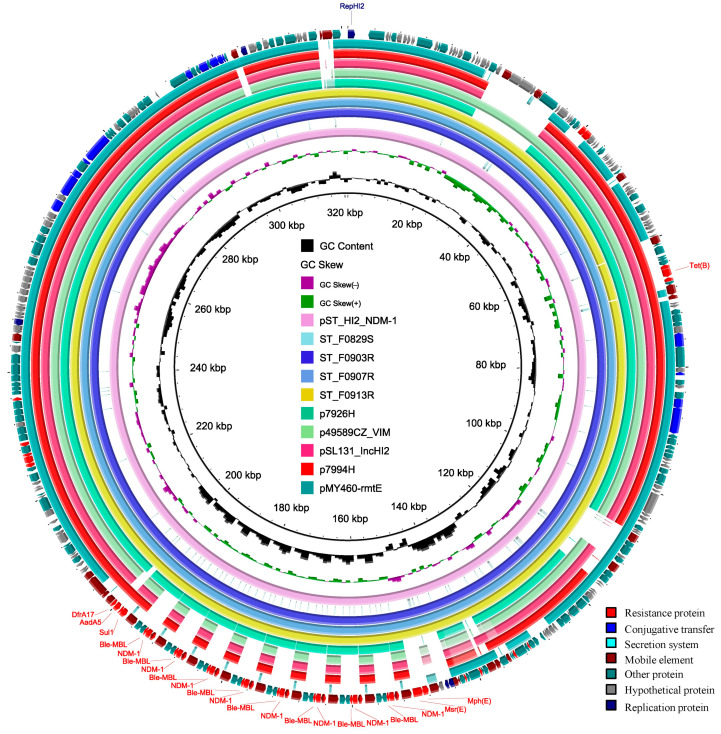
Genomic circle diagram of pST_HI2_NDM-1 and comparative genome analysis with pST_HI2_NDM-1 as the reference sequence. Circles range from 1 (the inner circle) to 13 (the outer circle). Circle 1, GC content inward indicates lower than the average GC content, and outward indicates higher than the average GC content; circle 2, GC skew (G − C/G + C), values > 0 are in green, and values < 0 are in purple; circle 4–7, the draft genomes of four *Salmonella* Typhimurium isolates. The outermost ring is the CDSs (encoding sequences), represented by the corresponding colored arrows.

**Figure 3 microorganisms-12-00020-f003:**
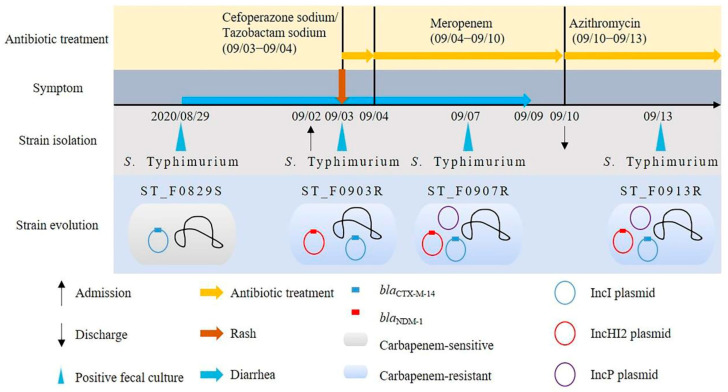
The medical history of the patient and the plasmid transfer events in the *Salmonella* Typhimurium isolate.

**Figure 4 microorganisms-12-00020-f004:**
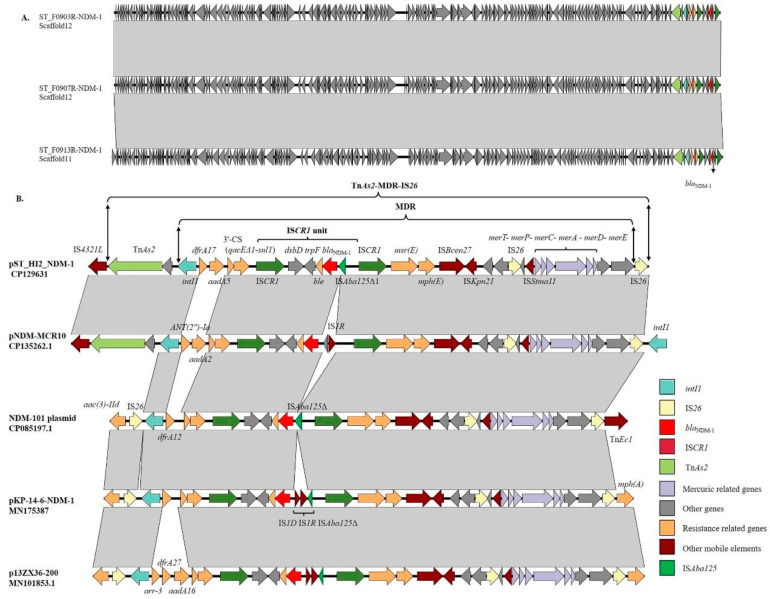
Genetic context of the *bla*_NDM-1_ gene and sequence alignment. (**A**). The sequence alignment of the *bla*_NDM-1_ gene in the draft genomes of ST_F0903R, ST_F0907R, and ST_F0913R isolates. (**B**). Genetic context of the *bla*_NDM-1_ gene on pST_HI2_NDM-1 and sequence alignment. The diagrams are to scale. The CDSs (encoding sequences) are represented by the corresponding colored arrows. Gray shading between sequences indicates the identity between the corresponding genetic loci.

**Figure 5 microorganisms-12-00020-f005:**
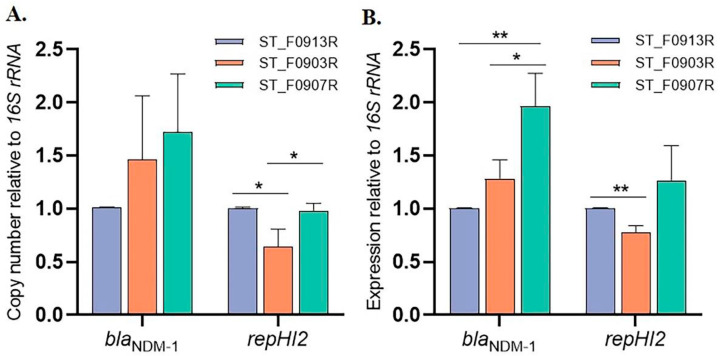
The real-time quantitative PCR (qRT-PCR) results. (**A**). The relative copy numbers of *bla*_NDM-1_ and IncHI-type plasmid replication initiator gene *repHI2*. (**B**). The relative expression levels of *bla*_NDM-1_ and IncHI-type plasmid replication initiator gene *repHI2*. The *16S rRNA* was used as the reference gene and the isolate ST_F0913R as the control group. * *p* < 0.05, ** *p* < 0.01.

**Table 1 microorganisms-12-00020-t001:** The MICs of *Salmonella* Typhimurium and transconjugants to antibiotics from the broth microdilution method.

Drug	MIC (mg/L)						
	ST_F0829S	ST_F0903R	ST_F0907R	ST_F0913R	ST_F0903Rtrans	ST_F0907Rtrans	ST_F0913Rtrans
IPM	0.125	16	16	16	16	16	16
IMR	0.25	16	16	16	16	16	16
MEM	≤0.06	64	128	64	64	128	64
MEV	≤0.06	128	128	64	128	128	64
CAZ	4	>128	>128	>128	>128	>128	>128
CZA	0.5	>128	>128	>128	>128	>128	>128
COL	1	1	1	1	1	1	1
POL	0.5	0.5	0.5	0.5	0.5	0.5	0.5
AMK	1	1	1	1	1	1	1
FEP	16	>128	>128	>128	>128	>128	>128
ATM	16	16	16	16	16	16	16
CIP	≤0.06	≤0.06	≤0.06	≤0.06	≤0.06	≤0.06	≤0.06
AZA	0.125	0.125	0.125	0.125	0.125	0.125	0.125
ERA	1	1	1	1	0.5	0.5	0.5
SXT	0.125	>128	>128	>128	>128	>128	>128
TGC	2	2	2	2	0.25	0.25	0.25

Abbreviations: minimum inhibitory concentration, MIC; Imipenem, IPM; Imipenem-relebactam, IMR; Meropenem, MEM; Meropenem-vaborbactam, MEV; Ceftazidime, CAZ; Ceftazidime/Avibactam, CZA; colistin, COL; Polymyxin, POL; Amikacin, AMK; Cefepime, FEP; Aztreonam, ATM; Ciprofloxacin, CIP; Amtreonam/Avibatan, AZA; Eravacycline, ERA; Trimethoprim/sulfamethoxazole, SXT; Tigecycline, TGC.

## Data Availability

All sequencing reads and assemblies were deposited in GenBank under the BioProject number PRJNA953093.

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
