# Peer review of "Salmonella Typhimurium with Eight Tandem Copies of blaNDM-1 on a HI2 Plasmid"

_microorganisms, 2023, doi:10.3390/microorganisms12010020_

Round 1

Reviewer 1 Report

Comments and Suggestions for Authors

This study presents an interesting case of Salmonella Typhimurium infections in a young patient in China. It is notable that while the first isolate recovered was susceptible to carbapenems, subsequent isolates displayed resistance due to the acquisition of an IncHI2 plasmid harboring the blaNDM-1 gene. The report effectively illustrates the dissemination of blaNDM-1 carrying plasmids among Salmonella species. However, I find the lack of discussion on why eight copies of blaNDM-1 did not confer higher carbapenem resistance compared to strains with a single copy of the gene to be a significant omission. This aspect is particularly intriguing as it represents the primary novelty of this study. Literature suggests that tandem copies of carbapenemase genes, such as blaVIM-1, can increase beta-lactam MICs, making this an area worthy of further exploration.

Specific Comments:

1.         Line 49: Please clarify whether it should be “are not further” or “were not further…” to maintain tense consistency.

2.         Line 72: Kindly add a reference for the EUCAST standards mentioned.

3.         Lines 311-312: Could the authors delve into the phenomenon of why multiple blaNDM-1 copies do not increase resistance? It would be beneficial to explore whether there is a difference in expression or production of blaNDM-1.

4.         Line 324: I suggest that the authors review the literature on IncHI2 plasmids. It is my understanding that these plasmids are known to carry other resistance genes, which could be relevant to your study (PMID 34451418, 22481058, 37335912 for example).

Author Response

Dear Reviewer,

Thank you for your kind consideration of our manuscript entitled “Salmonella Typhimurium with eight tandem copies of blaNDM-1 on a HI2 plasmid”. We have made corresponding revisions according to your suggestions. For convenience, the important amendments made in the original version of the manuscript were highlighted in red in the revised version.

Point-by-Point Response to Reviewers' Comments:

1.Line 49: Please clarify whether it should be “are not further” or “were not further…” to maintain tense consistency.

Response: It should be “were not further…”. We have revised the text accordingly.

2.Line 72: Kindly add a reference for the EUCAST standards mentioned.

Response: Thank you for your suggestion. We added a reference for the EUCAST standards.

The reference is as follows:

European Committee on Antimicrobial Susceptibility Testing. EUCAST clinical breakpoint table. https://www.eucast.org/clinical_breakpoints/.

3.Lines 311-312: Could the authors delve into the phenomenon of why multiple blaNDM-1 copies do not increase resistance? It would be beneficial to explore whether there is a difference in the expression or production of blaNDM-1.

Response: Thank you for your question. We appreciate your insightful comment. In response to your suggestion, we performed the Real-time quantitative PCR (qRT-PCR) experiment. The qRT-PCR results showed that the ST_F0903R isolate exhibited a significantly higher relative copy number of the blaNDM-1 gene compared to the ST_F0907R and ST_F0913R isolates (P<0.05) (Figure 5A). However, there was no statistically significant difference in the relative expression level of blaNDM-1 gene among the three isolates (Figure 5B). By combining the results obtained in this study with relevant literature, we have explored why the presence of multiple blaNDM-1 copies in this study did not lead to increased drug resistance. We found that the only difference among the three carbapenem-resistant Salmonella isolates was the presence of tandem copies of the ISCR1 unit carrying the blaNDM-1 gene in the ST_F0903R isolate, but an increase in the copy numbers of the blaNDM-1 gene did not increase its expression levels, thus not enhancing the MIC values of the isolate. Based on the above results, we have made the necessary revisions in the Methods, Result (last paragraph), Discussion (second-to-last and last paragraph), and Figure 5. For convenience, the important amendments made in the original version of the manuscript were highlighted in red in the revised version.

Regrettably, due to constraints in terms of time and resources, we were unable to conduct further experiments to explore the relationship between the tandem copy of the blaNDM-1 gene and its expression level. We have provided additional clarification on the limitations of this study in the Discussion section (last paragraph). We apologize for any inconvenience this may cause and sincerely appreciate your understanding.

4.Line 324: I suggest that the authors review the literature on IncHI2 plasmids. It is my understanding that these plasmids are known to carry other resistance genes, which could be relevant to your study (PMID 34451418 IF: 3.7 Q2 , 22481058, 37335912 for example).

Response: Thank you for your suggestion and the literature provided. Previous reports indicated that IncHI2 plasmids also carry other resistance genes, including blaNDM-5 [32], blaVIM-1 [33], and blaMCR-1 [34]. In this study, we did not find other carbapenemase genes other than blaNDM-1 on the plasmid. We have revised the text accordingly in the Discussion section (paragraph 3). For convenience, important modifications are highlighted in red.

The added references are as follows:

32.Ma, Z.; Zeng, Z.; Liu, J.; Liu, C.; Pan, Y.; Zhang, Y.; Li, Y. Emergence of IncHI2 Plasmid-Harboring blaNDM-5 from Porcine Escherichia coli Isolates in Guangdong, China. Pathogens 2021, 10, doi:10.3390/pathogens10080954.

33.Coelho, A.; Piedra-Carrasco, N.; Bartolomé, R.; Quintero-Zarate, J.N.; Larrosa, N.; Cornejo-Sánchez, T.; Prats, G.; Garcillán-Barcia, M.P.; de la Cruz, F.; González-Lopéz, J.J. Role of IncHI2 plasmids harbouring blaVIM-1, blaCTX-M-9, aac(6')-Ib and qnrA genes in the spread of multiresistant Enterobacter cloacae and Klebsiella pneumoniae strains in different units at Hospital Vall d'Hebron, Barcelona, Spain. Int J Antimicrob Agents 2012, 39, 514-517, doi:10.1016/j.ijantimicag.2012.01.006.

34.Sun, J.; Li, X.P.; Fang, L.X.; Sun, R.Y.; He, Y.Z.; Lin, J.; Liao, X.P.; Feng, Y.; Liu, Y.H. Co-occurrence of mcr-1 in the chromosome and on an IncHI2 plasmid: persistence of colistin resistance in Escherichia coli. Int J Antimicrob Agents 2018, 51, 842-847, doi:10.1016/j.ijantimicag.2018.01.007.

Reviewer 2 Report

Comments and Suggestions for Authors

This study has been conducted using state-of-the-art methods, and the results are well presented and conclusions are supported by the results. 

I have just some minor issues that I feel should be addressed.

Abstract, line 13 ad throughout the manuscript: You use the term strain, but basically, I assume it is merely isolates. Although the results of the PFGE are not presented, my guess is that the isolates all have the same PFGE profile and therefore represent the same strain. I suggest to use the term isolate rather than strain.

Line 22: When you write "an increase in resistance" do you the mean increase in MIC ?

Line 31: With the latest revisions of the scheme, we now call it the White-Kauffmann-le Minor scheme (Kauffmann with 2xf and 2xn)

Line 32: Reference 1 is not an adequate reference here. Line 32: non-typhoidal (not capital letter N)

Line 33: Not sure I understand what you mean by "complicated" here.

Line 33-34: Are you sure, NTS is the most common cause of self-limiting gastroenteritis? In that case, I would liek to see a reference for that. It is certainly not the case in many countries, where probably Campylobacter or E. coli or perhaps norovirus or rotavirus is more common.

Line 84-85: Please, rephrase sentence so that it is presented in past tense

Line 143 onwards: Where are the PFGE results presented? It is important to know whether they have identical patterns. A small figure showing this could be introduced.

References: The reference list does not look right. There is a doubble numbering of all references. In addition: species names and gene names must be in italict. And the MDPI style is not followed: the author names are not written correctly, the journal names are not written correctly in many references. Check the jounal´s style guide and go through each reference again.

Reference 17 is totally wrong. Use the citation suggested in the standard

And I assume there must be an ethical declaration, a statement about funding and one about conflicts of interests.

Author Response

Dear Reviewer,

Thank you for your kind consideration of our manuscript entitled “Salmonella Typhimurium with eight tandem copies of blaNDM-1 on a HI2 plasmid”. We have made corresponding revisions according to your suggestions. For convenience, the important amendments made in the original version of the manuscript were highlighted in red in the revised version.

Point-by-Point Response to Reviewers' Comments:

1. Abstract, line 13 ad throughout the manuscript: You use the term strain, but basically, I assume it is merely isolates. Although the results of the PFGE are not presented, my guess is that the isolates all have the same PFGE profile and therefore represent the same strain. I suggest to use the term isolate rather than strain.

Response: Thank you for your correction. The PFGE results of the four isolates of S. Typhimurium were shown in Supplementary Figure 1. The XbaI-PFGE band profiles of these strains exhibited a high degree of similarity and therefore represented the same strain. We have added the XbaI-PFGE results to the Results section (paragraph 1). We have made the revisions based on the issues you raised.

2. Line 22: When you write "an increase in resistance" do you the mean increase in MIC ?

Response: Thank you for your question. We apologize for the confusion. An increase in resistance here means an increase in MIC. We have made the revisions to the manuscript.

3. Line 31: With the latest revisions of the scheme, we now call it the White-Kauffmann-le Minor scheme (Kauffmann with 2xf and 2xn)

Response: Thank you for your correction. We have made the revisions to the manuscript.

4. Line 32: Reference 1 is not an adequate reference here.

Response: Thank you for your suggestion. Reference 1 was replaced by the following reference:

Knodler, L.A.; Elfenbein, J.R. Salmonella enterica. Trends Microbiol 2019, 27, 964-965, doi:10.1016/j.tim.2019.05.002.

5. Line 32: non-typhoidal (not capital letter N)

Response: Thanks for the comment. It has been changed as suggested.

6. Line 33: Not sure I understand what you mean by "complicated" here.

Line 33-34: Are you sure, NTS is the most common cause of self-limiting gastroenteritis? In that case, I would liek to see a reference for that. It is certainly not the case in many countries, where probably Campylobacter or E. coli or perhaps norovirus or rotavirus is more common.

Response: We apologize for the confusion. Our original intention was to indicate that there can be a variety of clinical manifestations associated with non-typhoidal serovars of Salmonella enterica (NTS), but the most frequently observed symptom is self-limiting gastroenteritis. Thank you for pointing out the error. We have made corresponding modifications in the Introduction section (paragraph 1).

7. Line 84-85: Please, rephrase sentence so that it is presented in past tense

Response: Thank you for your correction. We have made the necessary revisions based on the issues you raised.

8. Line 143 onwards: Where are the PFGE results presented? It is important to know whether they have identical patterns. A small figure showing this could be introduced.

Response: Thank you for your suggestion. The PFGE results of the four isolates of S. Typhimurium were shown in Supplementary Figure 1. The XbaI-PFGE band profiles of these strains exhibited a high degree of similarity and therefore represented the same strain. We have added the XbaI-PFGE results to the Results section (paragraph 1).

9. References: The reference list does not look right. There is a doubble numbering of all references. In addition: species names and gene names must be in italict. And the MDPI style is not followed: the author names are not written correctly, the journal names are not written correctly in many references. Check the jounal´s style guide and go through each reference again.

Reference 17 is totally wrong. Use the citation suggested in the standard

Response: Thank you for your correction. We have made the necessary revisions based on the issues you raised, including standardizing the naming of all strains and genes in this article.

10. And I assume there must be an ethical declaration, a statement about funding and one about conflicts of interests.

Response: Thank you for bringing this point to our attention. We have added the ethical declaration, a funding statement, and one statement about conflicts of interest after the main text of the article.

Ethical Approval

This study was reviewed and approved by the Medical Ethics Committee of Tongji Medical College, Huazhong University of Science and Technology (Number: 2021S013).

Disclaimers

All authors declare no conflicts of interest.

Funding

This work was supported by grants from the Special Foundation for National Science and Technology Basic Research Program of China (no. 2019FY101206 and no. 2019FY101200).

Reviewer 3 Report

Comments and Suggestions for Authors

The manuscript submitted by Yue Wang and co-authors is devoted to the analysis of antibiotic-resistant strain(s) of Salmonella typhimurium

Several issues don't let me recommend this manuscript to be published in Microorganisms in its current state.

  1. What is the main question addressed by the research?
The manuscript submitted by Yue Wang and co-authors is devoted to the analysis of antibiotic-resistant strain(s) of Salmonella typhimurium

2. Do you consider the topic original or relevant in the field, and if so, why?
The article will probably be of interest to some of the readers.

3. What does it add to the subject area compared with other published material?
The article considers a particular case that may be of interest to some researchers.

4. What specific improvements could the authors consider regarding the methodology? 5. Are the conclusions consistent with the evidence and arguments presented and do they address the main question posed?
In addition to the current text, the Discussion chapter should debate the physiological significance of the results obtained in the manuscript (related to the blaNDM etc).

6. Are the references appropriate?
Yes

7. Please include any additional comments on the tables and figures.   1) In "S. typhimurium" both words should be written in italics and the second word in lowercase

2) Chapter 3 should be named "Results."

3) Statistical analysis should be applied to the data presented in Table 1 and subsequent

4) The Discussion chapter should debate the physiological significance of the results obtained in the manuscript, related to the blaNDM etc.

Author Response

Dear Reviewer,

Thank you for your kind consideration of our manuscript entitled “Salmonella Typhimurium with eight tandem copies of blaNDM-1 on a HI2 plasmid”. We have made corresponding revisions according to your suggestions. For convenience, the important amendments made in the original version of the manuscript were highlighted in red in the revised version.

1) In "S. typhimurium" both words should be written in italics and the second word in lowercase
Response: Referring to previous literature published by Microorganisms, we found that in "Salmonella Typhimurium", only "Salmonella" should be italicized, and "Typhimurium" should be written in capital letters. We have ensured that this formatting is consistently applied throughout the manuscript for accuracy and adherence to the established conventions.

The reference is as follows:

Napoleoni M, Villa L, Barco L, Lucarelli C, Tiengo A, Baggio G, Dionisi AM, Angellotti A, Ferretti E, Ruggeri S, Staffolani M, Rocchegiani E, Silenzi V, Morandi B, Blasi G. Monophasic Variant of Salmonella Typhimurium 4,[5],12:i:- (ACSSuGmTmpSxt Type) Outbreak in Central Italy Linked to the Consumption of a Roasted Pork Product (Porchetta). Microorganisms. 2023 Oct 15;11(10):2567.

2) Chapter 3 should be named "Results."

Response: Thank you for your correction. We have made the revision based on the issues you raised.

3) Statistical analysis should be applied to the data presented in Table 1 and subsequent
Response: Thank you for your suggestion. Considering that the majority of our experimental results were related to bioinformatics analyses of whole genome sequencing, statistical analysis is not required. Furthermore, Table 1 solely presented the minimum inhibitory concentration (MIC) of four Salmonella Typhimurium isolates and their transconjugants, and therefore, statistical analysis is not applicable. The primary purpose of this table is to display the MIC values. We conducted an independent sample t-test on the real-time quantitative PCR data.

4) The Discussion chapter should debate the physiological significance of the results obtained in the manuscript, related to the blaNDM etc.

Response: Thank you for your suggestion. We have made the following revisions to the manuscript in response to your feedback:

In the Discussion section (paragraph 1), we highlighted the importance of studying carbapenem-resistant Salmonella.

In the Discussion section (paragraph 2), we emphasized the novelty of the patient samples and sequencing methods used in this study.

In addition, we performed the Real-time quantitative PCR (qRT-PCR) experiment. The qRT-PCR results showed that the ST_F0903R isolate exhibited a significantly higher relative copy number of the blaNDM-1 gene compared to the ST_F0907R and ST_F0913R isolates (P<0.05) (Figure 5A). However, there was no statistically significant difference in the relative expression level of blaNDM-1 gene among the three isolates (Figure 5B). By combining the results obtained in this study with relevant literature, we have explored why the presence of multiple blaNDM-1 copies in this study did not lead to increased drug resistance. We found that the only difference among the three carbapenem-resistant Salmonella isolates was the presence of tandem copies of the ISCR1 unit carrying the blaNDM-1 gene in the ST_F0903R isolate, but an increase in the copy numbers of the blaNDM-1 gene did not increase its expression levels, thus not enhancing the MIC values of the isolate. Based on the above results, we have made the necessary revisions in the Methods, Result (last paragraph), Discussion (second-to-last and last paragraph), and Figure 5. For convenience, the important amendments made in the original version of the manuscript were highlighted in red in the revised version.

Regrettably, due to constraints in terms of time and resources, we were unable to conduct further experiments to explore the relationship between the tandem copy of the blaNDM-1 gene and its expression level. We have provided additional clarification on the limitations of this study in the Discussion section (last paragraph). We apologize for any inconvenience this may cause and sincerely appreciate your understanding.

Reviewer 4 Report

Comments and Suggestions for Authors

The manuscript presents a method of isolating Carbapenem-sensitive and resistant strains from patient fecal samples. The methods and description are fine. Some suggestions are:

1) Most of the methods of sequencing and others are standard. It would be important to establish novelty of your methods and approach. You could discuss new HTS methods of testing for resistant strains.  

2) You could discuss challenges in testing patient samples, and the latest approaches for these tests. This would contribute to the novelty element of your work.

3) One suggestion would be discuss new experimental techniques to quantify resistance, such as this article: https://pubs.acs.org/doi/abs/10.1021/acssensors.9b01031

Author Response

Dear Reviewer,

Thank you for your kind consideration of our manuscript entitled “Salmonella Typhimurium with eight tandem copies of blaNDM-1 on a HI2 plasmid”. We have made corresponding revisions according to your suggestions. For convenience, the important amendments made in the original version of the manuscript were highlighted in red in the revised version.

Point-by-Point Response to Reviewers' Comments:

1) Most of the methods of sequencing and others are standard. It would be important to establish novelty of your methods and approach. You could discuss new HTS methods of testing for resistant strains.

Response: Thank you for your insightful comments and valuable suggestions. We recognize the significant potential of these new experimental techniques in drug resistance research and agree that highlighting the novelty of our methods and approach is important.

As our research focuses on the drug resistance mechanisms and genome evolution of strains, we did not discuss new HTS methods of testing for resistant strains in the discussion section. However, we emphasized the novelty of our sequencing method in detecting drug-resistant strains and unique samples, based on your suggestions. We have added relevant content to the Discussion section (paragraph 2) of the revised manuscript to improve the quality and innovation of our research.

In the future, we plan to investigate new experimental techniques for testing resistant strains, such as new HTS methods and the combination of multiple detection methods. We believe this will further enhance our understanding of drug-resistance mechanisms and contribute to the development of more effective treatments for drug-resistant bacterial infections.

2) You could discuss challenges in testing patient samples, and the latest approaches for these tests. This would contribute to the novelty element of your work.

Response to: Thank you for your insightful comments and valuable suggestions. We have added relevant content to the Discussion section (paragraph 2) of the revised manuscript to improve the quality and innovation of our research.

The added relevant content in the Discussion section (paragraph 2) is as follows:

Compared to the majority of studies that focus solely on single carbapenem-resistant strains, our research involved the continuous isolation of both carbapenem-sensitive and carbapenem-resistant Salmonella Typhimurium strains from fecal samples of a single patient. Through comprehensive genomic studies, we were able to gain a more in-depth understanding of the transfer process of carbapenemase genes and plasmids in Salmonella. By employing state-of-the-art PacBio HiFi sequencing methods, we revealed the evolution process of drug resistance in Salmonella due to the acquisition of the HI2 plasmid carrying the blaNDM-1 gene, along with eight tandem copies of ISCR1 unit (ISCR1-dsbD-trpF-ble-blaNDM-1-ISAba125Δ1-sul1Δ1) on the HI2 plasmid. This sequencing method can provide more accurate and complete genome assembly results, which is of great significance for the study of complex genomes [23].

The added reference is as follows:

  1. Hepner, S.; Kuleshov, K.; Tooming-Kunderud, A.; Alig, N.; Gofton, A.; Casjens, S.; Rollins, R.E.; Dangel, A.; Mourkas, E.; Sheppard, S.K.; et al. A high fidelity approach to assembling the complex Borrelia genome. BMC Genomics 2023, 24, 401, doi:10.1186/s12864-023-09500-4.

3) One suggestion would be discuss new experimental techniques to quantify resistance, such as this article: https://pubs.acs.org/doi/abs/10.1021/acssensors.9b01031 IF: 8.9 Q1

Response to: Thank you for your insightful comments and valuable suggestions. We appreciate the article you have listed, which introduces a new experimental technique combining microfluidic platforms with genetic methods like CRISPRi to uncover new genetic determinants of antibiotic susceptibility and evaluate the long-term effectiveness of antibiotics in bacterial cultures. We recognize the significant potential of these new experimental techniques in drug resistance research. As our research focuses on the drug resistance mechanisms and genome evolution of strains, we did not discuss new experimental techniques for quantifying drug resistance in the discussion section.

In the future, we plan to investigate new experimental techniques for testing resistant strains, such as new HTS methods and the combination of multiple detection methods. We believe this will further enhance our understanding of drug-resistance mechanisms and contribute to the development of more effective treatments for drug-resistant bacterial infections.

Round 2

Reviewer 1 Report

Comments and Suggestions for Authors

Although the isolate carrying 8 copies of blaNDM-1 remains as resistant as the other isolate, which is unusual, I commend the authors for performing copy number experiments. It would have been interesting to evaluate the plasmid copy number in parallel to determine if the same level of resistance could be explained by a lower plasmid copy number in the ST_F0903R isolate compared to the others. If the authors have this data, it would be valuable to include it in the study.

Comments on the Quality of English Language

Minor comments:

L162: Please rephrase “Copy numbers and expression levels of the blaNDM-1 gene were determined using qRT-PCR”

L166: Please correct “16Sr RNA” by “16S rRNA”

Author Response

Dear Reviewer,

Thank you for your kind consideration of our manuscript entitled “Salmonella Typhimurium with eight tandem copies of blaNDM-1 on a HI2 plasmid”. We have made corresponding revisions according to your suggestions. For convenience, the important amendments made in the original version of the manuscript were highlighted in red in the revised version.

Point-by-Point Response to Reviewers' Comments:

Reviewer #1:

Although the isolate carrying 8 copies of blaNDM-1 remains as resistant as the other isolate, which is unusual, I commend the authors for performing copy number experiments. It would have been interesting to evaluate the plasmid copy number in parallel to determine if the same level of resistance could be explained by a lower plasmid copy number in the ST_F0903R isolate compared to the others. If the authors have this data, it would be valuable to include it in the study.

Response: Thank you for your insightful comments and valuable suggestions. In response to your suggestion, we conducted a real-time quantitative PCR (qRT-PCR) experiment again. The qRT-PCR results revealed that, when using the 16S rRNA gene as a reference, there was no significant difference in the relative copy number of blaNDM-1 among the three isolates (Figure 5A). However, ST_F0907R exhibited a significantly higher relative expression level of blaNDM-1 compared to the other two isolates (Figure 5B). Moreover, the relative copy number and expression level of repHI2 in ST_F0903R were significantly lower (Figure 5A, 5B). Based on the above results, we have made the necessary revisions in the Methods, Result (last paragraph), Discussion (second-to-last paragraph), and Figure 5. For convenience, the important amendments made in the original version of the manuscript were highlighted in red in the revised version.

The added content in the Discussion section (second-to-last paragraph) is as follows:

Notably, our research findings indicated that the only difference in the IncHI2 plasmid among the three carbapenem-resistant Salmonella isolates was the copy number of the ISCR1 unit carrying blaNDM-1. Despite the presence of multiple copies of blaNDM-1 on the IncHI2 plasmid in ST_F0903R, the copy number of this plasmid was lower compared to the other isolates. As a result, this did not lead to an increase in the expression level of blaNDM-1, thus explaining the similar levels of resistance observed in the three isolates.

Minor comments:

L162: Please rephrase “Copy numbers and expression levels of the blaNDM-1 gene were determined using qRT-PCR”

Response: Thank you for your suggestion. We have made the revisions based on the issues you raised.

L166: Please correct “16Sr RNA” by “16S rRNA”

Response: Thank you for your correction. We have made the revisions based on the issues you raised.

Reviewer 3 Report

Comments and Suggestions for Authors

The authors have significantly changed the manuscript, but some issues should be resolved before the paper might be recommended for publication.

1) The version of the paper that was sent after R1, doesn't have any Figures, and the reviewer cannot be sure that no changes were made to the Figures. 

2) The Introduction section is extremely concise in its current state. It would be nice to add some paragraphs about blaNDM-1 2 and the HI2 plasmid

3) "Materials and methods" section should contain information about how the strain of Salmonella Typhimurium was obtained (patient information, medical ethical, etc). 

Sincerely,

Author Response

Dear Reviewer,

Thank you for your kind consideration of our manuscript entitled “Salmonella Typhimurium with eight tandem copies of blaNDM-1 on a HI2 plasmid”. We have made corresponding revisions according to your suggestions. For convenience, the important amendments made in the original version of the manuscript were highlighted in red in the revised version.

Point-by-Point Response to Reviewers' Comments:

Reviewer #3:

1) The version of the paper that was sent after R1, doesn't have any Figures, and the reviewer cannot be sure that no changes were made to the Figures.

Response: Thank you for your reminder. I apologize for the oversight in the previous revision, where only Figure 5 was uploaded. I understand the inconvenience caused and assure you that this time I have re-uploaded all Figures in the manuscript.

2) The Introduction section is extremely concise in its current state. It would be nice to add some paragraphs about blaNDM-1 2 and the HI2 plasmid

Response: Thank you for your suggestion. We added some content about blaNDM-1 and HI2 plasmids in the Introduction section (second paragraph) and made necessary modifications to the references.

The added content is as follows:

Since it was first reported in 2009 [11], the blaNDM-1 gene-positive Enterobacterales has spread rapidly around the world [12]. The blaNDM genes were usually reported to be located on the IncX3, IncC, IncL, IncM, and IncN plasmids [12]. The IncHI2 plasmid carrying blaNDM genes is rarely reported [12-14]. Recent research has shown that mobile genetic elements (MGEs) play a crucial role in facilitating the rapid transmission of blaNDM genes [15,16]. Such MGEs, like IS26 and ISCR1, which are frequently found in the vicinity of blaNDM genes among diverse strains, most likely contribute to the dissemination of blaNDM genes [17,18].

3) "Materials and methods" section should contain information about how the strain of Salmonella Typhimurium was obtained (patient information, medical ethical, etc).

Response: Thank you for your suggestion. We have added information on how Salmonella typhimurium strains were obtained and case information in the Materials and Methods section.

The added content is as follows:

2.1. Bacterial isolates and case information

A total of four isolates used in this study were derived from fecal specimens on different dates obtained from an 11-month-old patient who was admitted to Tongji Hospital: August 29th (ST_F0829S), September 3rd (ST_F0903R), September 7th (ST_F0907R), and September 13th (ST_F0913R), 2020. The patient presented with symptoms of diarrhea (4-5 times/day) and fever upon admission. Cefoperazone-tazobactam was administered on September 2, but the patient developed a wind-like rash on the face after treatment. After oral administration of loratadine on September 3, the patient's rash subsided, indicating a possible allergic reaction of the patient to Cefoperazone-tazobactam. Consequently, the medication was switched to meropenem on September 4. However, on September 10, the patient's diarrhea symptoms did not significantly improve. Therefore, meropenem was discontinued, and azithromycin was administered instead. After the azithromycin treatment, the patient's condition improved.

Additionally, the ethical declaration is placed at the end of the manuscript as follows:

Ethical Approval: The data produced in this study was not used for the treatment or management of patients, therefore informed patient consent was not required. This study was reviewed and approved by the Medical Ethics Committee of Tongji Medical College, Huazhong University of Science and Technology (Number: 2021S013).